# A Retrospective Study on tDCS Treatment in Patients with Drug-Resistant Chronic Pain

**DOI:** 10.3390/biomedicines12010115

**Published:** 2024-01-05

**Authors:** Yolanda A. Pérez-Borrego, Vanesa Soto-León, Ángela Brocalero-Camacho, Antonio Oliviero, Carmen Carrasco-López

**Affiliations:** 1FENNSI Group, Hospital Nacional de Parapléjicos, SESCAM, 45071 Toledo, Spain; vsleon@sescam.jccm.es (V.S.-L.); abrocalero@sescam.jccm.es (Á.B.-C.); antonioo@sescam.jccm.es (A.O.); 2Unidad de Neurología, Hospital de Parapléjicos, SESCAM, 45071 Toledo, Spain; 3Hospital Los Madroños, 28690 Brunete, Spain; 4Internet of Things and People, University of Malmö, 211 19 Malmö, Sweden; 5Department of Anatomy, University of Seville, 41009 Seville, Spain

**Keywords:** chronic pain, tDCS, NIBS, pain intensity, motor cortex

## Abstract

**Background**. Transcranial direct current stimulation (tDCS) of the primary motor cortex (M1) has an analgesic effect superior to a placebo in chronic pain. Some years ago, tDCS was implemented at the Hospital Nacional of Paraplegics (Toledo, Spain) to treat patients with pharmacological resistance to chronic pain. **Objective**. The main objectives of this study with tDCS were (1) to confirm the safety of one-year treatment; (2) to estimate the number of patients after one year in treatment; (3) to describe the effects of tDCS on the pain intensity during one-year treatment; and (4) to identify factors related to treatment success. **Methods**. This was a retrospective study conducted at the National Hospital for Paraplegics with 155 patients with pharmacologically resistant chronic pain. Anodal tDCS was applied over the M1 for 20 min at 1.5 mA for 10 treatment sessions from Monday to Friday (*Induction phase*), followed by 2–3 sessions per month (*Maintenance phase*). Pain intensity was assessed using a Visual Analogue Scale (VAS). **Results**. Anodal tDCS on M1 confirmed the reduction in the pain intensity. Moreover, 58% of outpatients completed one year of treatment. Only the VAS values obtained during the baseline influenced the response to treatment. Patients with a very high VAS at the baseline were more likely to not respond adequately to tDCS treatment. **Conclusions**. Anodal tDCS over M1 is an adequate therapy (safe and efficient) to treat drug-resistant chronic pain. Moreover, pain intensity at the start of treatment could be a predictor of patients’ continuity with tDCS for at least one year.

## 1. Introduction

Pain is defined as an unpleasant sensory and emotional experience associated with actual or potential tissue damage or described in terms of such damage [1]. Usually, pain is regarded as chronic when it lasts or recurs for more than 3 to 6 months [2]. Pain caused by damage to the central nervous system (CNS) or peripheral nervous system (PNS) is defined as neuropathic pain (NP) [3]. The etiologies of nervous system pain disorders are numerous and different. For instance, infections, nutritional deficiencies, multiple sclerosis and stroke, to mention a few, can cause central NP [4]. Fibromyalgia (FM) is a chronic pathology, characterized by generalized musculoskeletal pain and a decreased pressure pain threshold [5,6,7]. Pharmacological treatment is the most common therapeutic strategy to manage acute and chronic pain. Especially in chronic pain management, pharmacological treatment often presents disadvantages such as habituation, side effects and a lack of efficacy. Thus, an important number of individuals suffering from chronic pain can be considered as drug-resistant patients. In these cases, transcranial direct current stimulation (tDCS) represents a promising and safe complement to medication. In chronic NP, the most used target for tDCS is the motor cortex (M1) [8]. A recent evidence-based guidelines paper recommended tDCS over M1 in several chronic NP syndromes [9]. tDCS has numerous advantages as a beneficial clinical tool: it is safe and not excessively expensive and professionals can be trained in a brief period [8]. Additionally, it can be proposed for at-home treatment [10,11].

In recent years, tDCS has become a common clinical practice at the Hospital Nacional of Paraplegics in Spain. The therapeutic intervention requires repeated tDCS daily sessions (often 10 sessions from Friday to Monday) [12,13,14]. This period is called the “*induction phase*”. After the induction phase, the number of tDCS sessions is reduced to 2–3 per month. This period is named the “*maintenance phase*”. All the patients in this study suffered from drug-resistant chronic pain (most of them had a neuropathic component of pain including FM). It has been shown that real tDCS presents an analgesic effect superior to the placebo effect obtained using sham tDCS [9]. On the other hand, since not all patients treated equally benefit from tDCS treatment, a number of patients are considered “*non-responders*” to tDCS or, more generally, to non-invasive brain stimulation (NIBS) techniques. The definition of “*responder*” in pain medicine is not trivial. A reduction in pain intensity of a given value (e.g., 30% or 50% of the VAS, where “VAS” denotes the Visual Analogue Scale), a self-reported meaningful improvement and a reduction in the use of painkillers and other strategies can be used to define the “*responder*” [12,13]. Hitherto, due to the fluctuations in pain intensity and the emotional state of patients, it is often difficult to decide whether a patient is a “*responder*” based on the mere evaluation of such variables.

Notably, it is considered that for long-term treatment (one year of treatment), a valuable aspect is the willingness of the patient to follow the treatment. Patient commitment is thus a direct measure of the clinical benefit that takes into account the reduction of pain, the discomfort caused by the logistics (e.g., transportation, visit the hospital, waiting periods, etc.) and the presence of side effects (discomfort caused by the procedure). Patient engagement is thus key in clinical trials testing analgesia [15]. For this reason, we consider as “*responders*” all patients that are still receiving treatment at up to one year. Moreover, we “quantify” the pain reduction (if any) after a long-term period (one year) by the VAS scale.

The main objectives of this study were (1) to confirm the safety and efficiency of one-year treatment with tDCS; (2) to calculate the number of “*responder*” patients to one-year treatment with tDCS; (3) to describe the effects of tDCS on pain intensity during one-year treatment; and (4) to identify factors related to the success of tDCS treatment.

## 2. Materials and Methods

### 2.1. Subjects

Subjects with drug-resistant chronic pain were referred to the FENNSI group from different departments of the National Hospital for Paraplegics and from other hospitals in Spain (mainly from pain units and neurology and rehabilitation departments). Drug-resistant chronic pain refers to the fact that pharmacological treatment was insufficient to reduce pain due to a lack of efficacy or habituation, and even risks of addiction, especially when considering opioids [14]. Outpatients with reduced drug tolerance due to drug side effects were included in the study. All the subjects were outpatients with drug-resistant chronic pain derived for treatment with tDCS.

Data from 161 individuals with drug-resistant chronic pain were considered for this study; 3.7% of them were discarded as they had missing or incomplete data. Ultimately, 155 outpatients were included in the final analysis (demographic and clinical characteristics are reported in Table 1). Concomitant medication was allowed (including painkillers) and was kept stable according to the guidelines of the referring physicians (Table 1). All the patients signed an informed consent form. The protocol is included in our clinical practice and was approved by the hospital management and the local ethics committee (“Hospital Universitario de Toledo”; reference number 56; date 18 October 2006).

The outpatients were considered suitable for treatment if they met the following criteria: (1) drug-resistant chronic pain or drug intolerance over 3 months, with clinical stability; (2) chronic non-degenerative pathology and no rapidly progressive diseases; and (3) severe or moderate pain. The exclusion criteria were (1) age less than 18 or more than 75 years old; (2) metal objects in the head; (3) progressive or neurodegenerative disease; (4) major psychiatric disorder; (5) changes in pharmacological treatment 3 months before starting the tDCS treatment; (6) short-term major pharmacological changes foreseeable; and (7) dementia or cognitive disorder.

Patients with pain localized in one hemi-body were treated by stimulating the contralateral M1, whilst patients with bilateral pain or in which the underling disorders may affect bilateral regions (e.g., SCI) were simultaneously stimulated in both M1 regions. More details of the protocols will be provided in the Section 2.

### 2.2. Transcranial Direct Current Stimulation and Placement of Electrodes

tDCS was delivered by a transcranial direct current stimulator (HDC stim, Newronika^®^, Milan, Italy). The stimulation period during the induction phase took place for 10 consecutive days (two weeks, Monday–Friday). Regarding the parameters of anodal tDCS stimulation on M1 [12,13,14], the anode electrodes were placed over C3 or C4 (unilateral) or both (bilateral) following the 10/20 International System for electroencephalography (EEG), with a saline-soaked pair of surface electrodes (25 cm^2^). On the other hand, the cathode electrode was placed on the contralateral supraorbital area (25 cm^2^ if unilateral) or in the arm (50 cm^2^ if bilateral stimulation) [9]. The stimulation had an intensity of 1.5 mA for 20 min. The physician decided the montage of the electrodes during the recruitment phase, as unilateral or bilateral, depending on the pain location. For patients with pain located in one side of the body, the contralateral M1 was stimulated [16].

Regarding the tDCS’ security, the profile is very high based on evidence [17]. The adverse effects are usually local sensory discomfort such as itching, tingling, a burning sensation under the electrodes and mild headache [18,19,20]. In general, it is important to avoid local skin burns by limiting excessive current density according to the size and the shape of the electrode and maintaining good contact between the electrodes and the skin [21,22]. All subjects were informed about the phenomena described above before starting the treatment.

### 2.3. Clinical Assessment

A physician confirmed the diagnosis and evaluated the inclusion and exclusion criteria. Moreover, an assessment of cognitive and emotional status was performed. The Mini-Mental Status Examination (MMSE) [23] was used to screen cognitive alterations, and the Beck Depression Inventory (BDI) [24] was used to detect and quantify depressive mood. At the end of the first visit, during the recruitment phase, the physician or neuropsychologist described carefully to the patient how to complete the Visual Analogue Scale (VAS) [25], explaining that the VAS represents how much pain they experience at the moment of completing the scale. Subjects had to make a vertical mark on a horizontal line of ten centimeters, with the left side indicating the value of 0, or no pain, and the right indicating a value of 10, or the maximum pain that they could imagine. Afterwards, we provided 5 VAS scales to assess pain once per day prior to the induction phase. The Patient Global Impression of Changes (PGIC) [26] reflected the patient’s impressions of the efficacy of the tDCS treatment. One week after the last tDCS induction session, a phone call was made, and patients were asked about their impressions of any changes, either positive or negative, related to pain. This used a Likert-type scale with 7 different options to choose from: 1 = very much improved; 2 = much improved; 3 = minimally improved; 4 = no change; 5 = worse; 6 = much worse; 7 = very much worse.

### 2.4. Phases of the Protocol and Experimental Design

This was a retrospective study without a sham comparator. The protocol of the present study was derived from the literature [9] and from previous experience. It consisted of three main phases: (1) *recruitment*; (2) *induction*; and (3) *maintenance* (Figure 1).

*1. Recruitment phase:* In the recruitment phase, a physician made an assessment and decided whether the patient fit the inclusion/exclusion criteria for tDCS treatment. Then, a neuropsychologist administered the MMSE and BDI. The patients were asked to fill out a VAS for a 5-day period (once a day) at home before the induction phase started, to record their pain intensity. The mean value of the 5 days was used for the analysis (*VAS_Baseline*).

*2. Induction phase:* This was the first intervention period with tDCS. During this period, subjects were stimulated by tDCS, at the hospital, for 20 min at 1.5 mA of electrical induction, for 10 days (2 weeks, Mon–Fri). Subjects were placed in a comfortable chair and always in the same environment. One week after the last tDCS induction session, a phone call was made to the patients to ask them about their impression of any changes, positive or negative, related to the pain using the PGIC [26]. The *pain intensity* with the VAS was assessed over time at 5 days (T0).

*3. Maintenance phase:* Outpatients came to the hospital every 15 days on average, to receive only one tDCS session (2–3 sessions per month). The parameters of the stimulation and the position were the same as in the induction phase. The *maintenance phase* was divided into T1, T2 and T3, corresponding to 3, 6 and 12 months. As a follow-up assessment, the *pain intensity* was always registered by the VAS before the stimulation in each session. The mean value of all sessions during each period defined previously was used for the analysis (*VAS_T0*, *VAS_T1*, *VAS_T2*, *VAS_T3*).

### 2.5. Statistical Analysis

The clinical and demographic characteristics of the patients were analyzed by descriptive statistics, showing the results by mean and standard deviation (SD) for parametric data and median and interquartile range (IQR) for non-parametric data. The Shapiro–Wilk test evaluated the variables’ normality. A two-tailed binomial test was used to compare the proportions of women to men in the study. Parametric univariate tests (one-way ANOVA) and nonparametric univariate tests (Chi-squared test and Kruskal–Wallis test) were used to compare demographics and clinical data between the different *diagnostic groups* of patients. The group of patients who continued tDCS treatment for 12 months (T3) was considered the “*responder*” group. In this group of patients, we analyzed the evolution of the *pain intensity* over time (*VAS_Baseline*, *VAS_T0*, *VAS_T1*, *VAS_T2*, *VAS_T3*) with a Friedman test. In the case of significant effects, Conover’s post hoc analysis was performed to identify the difference. In addition, we performed an analysis with the Chi-square test to compare the number of patients who dropped out of the treatment at the different time points of the study (T0, T1, T2) with the *type of stimulation* (bilateral or unilateral).

Moreover, a logistic regression was performed to determine the effects of the independent *variables of age*, *sex*, *duration of pain* and *VAS_Baseline* on the probability of maintaining stimulation treatment for 12 months. Spearman’s correlation test was performed to study first the correlation between *PGIC* and the continuity of treatment and second between the *PGIC* and *VAS_Baseline*.

For the analysis of the data, the JASP version 0.16.1 software was used. The results were considered significant at *p* < 0.05.

## 3. Results

### 3.1. Description of the Sample

A report of the clinical and demographic characteristics of the patients, including medication at baseline, is described in Table 1. The total number of subjects included in the analysis was 155 (mean age: 49.26 ± 11.69 years), with a significantly higher representation of women (61.3% vs. 38.7%). *VAS_Baseline* was considered as the VAS collected during the 5 days before the *induction phase*. Fifty percent of patients had a *VAS_Baseline* greater than 76.0 mm with a *duration of pain* longer than 5 years. The most prescribed drugs for patients included in the study were antidepressants (58.1%). Ninety-eight percent of patients had normal MMSE values (MMSE: 25–30), classified as having no cognitive impairment, and only 2% had MMSE values defined as mild cognitive impairment (MMSE: 20–24). Regarding BDI values, patients were classified as *not depressed* (21%, BDI: 0–10), *intermittent moods* (23%, BDI: 11–16), *mild mood disturbance* (12%, BDI: 17–20), *moderately depressed* (28%, BDI: 21–30) and *severely depressed* (16%, BDI: >31). Next, patients were categorized into four different *diagnostic groups*: SCI (N = 51; 32.9%), FBSS (N = 15; 9.7%), FM (N = 40; 25.8%) and others, such as post-herpetic neuralgia, trigeminal neuralgia, diabetic neuropathy, radiculitis, etc. (N = 49; 31.6%). The demographic characteristics of the patients according to each *diagnostic group* are described in Table 2.

There was a significant difference between the *diagnostic groups* with respect to *gender* (Chi-square test: χ^2^ = 36.178; *p* < 0.001), with females being more represented in the FM (92.5%) and FBSS (66.7%) groups and males more represented in the SCI group (68.6%). In contrast, there was no significant difference in *age* with respect to *diagnostic groups* (one-way ANOVA: F(3) = 0.36; *p* = 0.78). Furthermore, no significant difference was found in *VAS_Baseline* (Kruskal–Wallis test: H(3) = 5.04; *p* = 0.16) and MMSE (Kruskal–Wallis test: H(3) = 2.51; *p* = 0.47) among the *diagnostic groups*. On the other hand, when we compared the different *diagnostic groups*, we found a significant difference in the *duration of pain* (Kruskal–Wallis test: H(3) = 18.05; *p* < 0.001) and in *BDI* (Kruskal–Wallis test: H(3) = 13.68; *p* < 0.003). More specifically, the FM group had a longer *duration of pain* (10 [6–17] years) and was more depressed, with the highest BDI values (26 [16–34]).

### 3.2. Description of the Treatment Effects

#### 3.2.1. Safety and Discomfort Caused by tDCS

In our study, patients performed around 46 tDCS sessions in one year, and none of the outpatients mentioned any severe adverse effects. Subjects felt the current as an itching sensation at the electrode site. Moreover, some individuals reported a metallic taste during the tDCS session. No severe adverse events were reported. On the contrary, some individuals reported that they felt relaxed during the tDCS session.

#### 3.2.2. “Responders”

Patients that successfully continued the treatment for up to 12 months were categorized as “*responders*” to tDCS. Fifty-eight percent (N = 90) of them were “*responders*”. The number of drop-out patients (“*non-responders*”) was lower than 20% for each time point (14.83% at T0; 16.12% at T1; 10.96% at T2). The percentages of “*responders*” in each *diagnostic group* were (1) SCI: 25.5%; (2) FM: 32.2%; (3) FBSS: 8.9%; and (4) others: 33.4%.

#### 3.2.3. Influence of Type of Stimulation

According to the *type of stimulation*, unilateral tDCS was applied in 14 patients (9.03%) and bilateral tDCS was administrated in 141 patients (90.97%). To analyze whether the *type of stimulation* had an impact on the number of patients that dropped out (“*non-responders*”) at each time point in the study (T0, T1, T2), we performed a *Chi-square* test. The analysis showed no difference between the *type of stimulation* groups (Chi-square test: χ^2^ = 0.34; *p* = 0.95). To further confirm whether the *type of stimulation* affected the *pain intensity* in the “*responder*” group at T3, we performed a Mann–Whitney *t*-test and we found no difference (*p* = 0.73). In consequence, we pooled our data within the *type of stimulation* category.

#### 3.2.4. tDCS Effects on Pain Intensity in the “Responder” Group

The reduction in *pain intensity* in the “*responder*” group compared to the *VAS_Baseline* was 15.2% at T0; 26.9% at T1; 24.5% at T2; and 26.4% at T3. The Friedman test analysis showed that the *pain intensity* in the “*responder*” group was significantly reduced over time (χ^2^ = 98.69, *p* < 0.001; Conover’s post hoc: *VAS_Baseline* vs. *VAS_T0*, *p* < 0.001; *VAS_Baseline* vs. *VAS_T1*, *p* < 0.001; *VAS_Baseline* vs. *VAS_T2*, *p* < 0.001; *VAS_Baseline* vs. *VAS_T3*, *p* < 0.001) (Figure 2). When we compared the *VAS_Baseline* vs. *VAS_T3* in the “*responder*” group separated according to *diagnostic groups*, there was a reduction in *pain intensity* as follows: (1) SCI: 37.6%; (2) FM: 10.6%; (3) FBSS: 27.8%; (4) others: 25.5%. These data showed that the analgesic effect in the “*responder*” group increased in the first three months and then remained stable.

#### 3.2.5. Probability to Predict the “Responders” at Baseline Evaluation

Logistic regression was performed to determine the effects of the independent variables of *age*, *sex*, *duration of pain* and *VAS_Baseline* on the probability of being a “*responder*”. The logistic regression model was significantly described by the variables (χ^2^(150) = 9.880; *p* = 0.042), although with a weak ability to explain across variables whether a subject would continue the treatment for 12 months (Nagelkerke, R^2^ = 0.083). The model correctly classified 61.3% of cases with sensitivity of 80% and specificity of 35.4%, so this model was not considered to have strong predictive ability. The only significantly different variable in the model was *VAS_Baseline*. Lower *VAS_Baseline* values were associated with a higher probability of being a “*responder*” (*p* = 0.014; OR = 0.975; 95% CI:0.955–0.995). In light of these results, we decided to make a comparison between the group of “*responders*” and “*non-responders*” with the only significant variable in the model: *VAS_Baseline*. We confirmed that “*responders*” had a significantly lower *VAS_Baseline* value (75.50 [57–83.75]) than “*non-responders*” (80 [67–89]) (Mann–Whitney test = 3543; *p* = 0.025). On the other hand, *VAS_Baseline* was not independent. As expected, *BDI* was strongly correlated in the regression analysis (rho = 0.260; *p* = 0.001), which is the reason that it was not included in the regression analysis.

#### 3.2.6. “Responders” and PGIC

*PGIC* was obtained one week after the induction phase. Spearman’s correlation test showed a negative perception of the patient regarding the treatment (high scores on PGIC) and it was related to treatment discontinuation (rho = −0.39; *p* = 0.002). We obtained that the 74.44% of the “*responders*” scored low values (1, 2 or 3), while 66.15% of the “*non-responders*” answered with high values (4, 5, 6 or 7). Moreover, when we associated the PGIC and the *VAS_Baseline*, Spearman’s correlation test revealed a positive correlation (rho = 0.2; *p* = 0.01). Altogether, these results suggest that patients having a positive response after *the induction phase* with less *pain intensity* at baseline are more likely to be “*responders*”.

## 4. Discussion

In the present study, we examined the long-term effects produced by tDCS sessions over M1 in 155 outpatients with drug-resistant chronic pain. The main results can be summarized as follows: (1) tDCS is safe; (2) the number of “*responders*” is approximately 60% of the treated patients; and (3) the maximal pain reduction is achieved after three months. We failed to find a clear feature allowing us to predict the “*responder*” and “*non-responder*” populations. Our results showed that only the pain intensity reported prior to the treatment determined the response to tDCS. Hence, patients with a very high VAS at baseline are more likely to not respond adequately to tDCS treatment.

### 4.1. Safety of the Treatment

In our study, patients who completed one year of treatment performed 46 sessions on average and none of the outpatients mentioned any severe adverse effects. Only some of them mentioned light to moderate sensations such as itching and a metallic taste during the stimulation, on the one hand, and redness on the skin after the treatment, on the other, as has been described in the literature [18,19,27,28,29,30]. The reported adverse effects were considered mild according to the recommendations of the *International Conference on Harmonisation* guidelines, requiring no medical treatment [31].

### 4.2. Efficacy of the Treatment

We confirmed the analgesic effects of motor cortex tDCS reported in the literature and guidelines [8,9,32]. One of our main objectives was to calculate the number of “*responders*” at 12 months of treatment with tDCS. Fifty-eight percent of outpatients continued with tDCS treatment for up to 12 months and were considered “*responders*”. Related to the long-term effects of tDCS on *pain intensity* in the “*responder*” *group*, the pain reduction was more evident after the *induction phase* (T0) and reached the maximum after 3 months (T1). Once the analgesic effect peaked at T1, there was no further increment. These results highlight the importance of maintaining tDCS treatment in order to reduce the pain intensity over time. A similar conclusion was reached by Brocalero-Camacho and colleagues in a study that showed how the pain intensity returned to baseline levels in patients with chronic neuropathic pain after the tDCS treatment was interrupted due to the COVID-19 lockdown [33]. We found that, at least in the first 3 months, a cumulative analgesic effect with repeated tDCS sessions was achieved. Next, when a plateau was reached, the repeated tDCS sessions allowed us to keep this effect stable over time, in agreement with another previous work [8]. It is important to emphasize that this was a retrospective study without a sham comparator, so real tDCS and placebo effects coexisted.

### 4.3. Treatment Success Factors

Our study revealed that the main factor related to the success of tDCS treatment was the level of pain prior to the stimulation. As such, patients with low values of VAS at baseline (*VAS_Baseline*) are more likely to be “*responders*”. On the other hand, when we compared the *VAS_Baseline* between the “*responder*” and “*non-responder*” groups, we found a difference of less than 7%. Such a modest difference does not allow us to use the *VAS_Baseline* variable as a predictive biomarker in clinical practice. Of note, no other variables showed consistent and robust predictive value. In consequence, further studies will be necessary to clarify whether the pain intensity at baseline can be associated with other variables to create a predictive model.

The PGIC obtained after the *10-day stimulation period* showed that patients with improvements were more likely to become “*responders*”. This subjective scale has been shown to be reliable and effective [34,35]. In 2001, Farrar and colleagues reported a consistent relationship between the change in pain intensity numerical rating scale and the PGIC, regardless of disease type, age, sex or treatment group [34]. The authors observed that a reduction of approximately 30% in the pain intensity numerical rating scale from baseline to post-treatment represented a clinically important difference. This relationship between the pain intensity numerical rating scale percentage of change and the PGIC was also coherent regardless of baseline pain. Our results indicate the same direction, with a positive correlation between *VAS_Baseline* and the PGIC, indicating the lower success of the treatment with higher values of *VAS_Baseline*.

### 4.4. Limitations

The first limitation of our study is related to the heterogeneous sample (e.g., diagnoses and locations of pain were different). Second, patients were under different pharmacological treatments. We instructed the patients not to interrupt their medication during the tDCS treatment. Finally, our sample was not large enough to evaluate whether tDCS interfered (positively or negatively) with the drug profile of each patient.

## 5. Conclusions

Our results showed that only the pain intensity reported prior to the treatment could determine the response to tDCS. In this study, the number of patients who continued with tDCS treatment for one year was approximately 60% and the maximal pain reduction was achieved after three months. Thus, tDCS has several advantages to be used as a beneficial clinical tool to treat drug-resistant chronic pain: it is safe and economical, and clinicians and researchers can be trained in a short period of time.

## Figures and Tables

**Figure 1 biomedicines-12-00115-f001:**
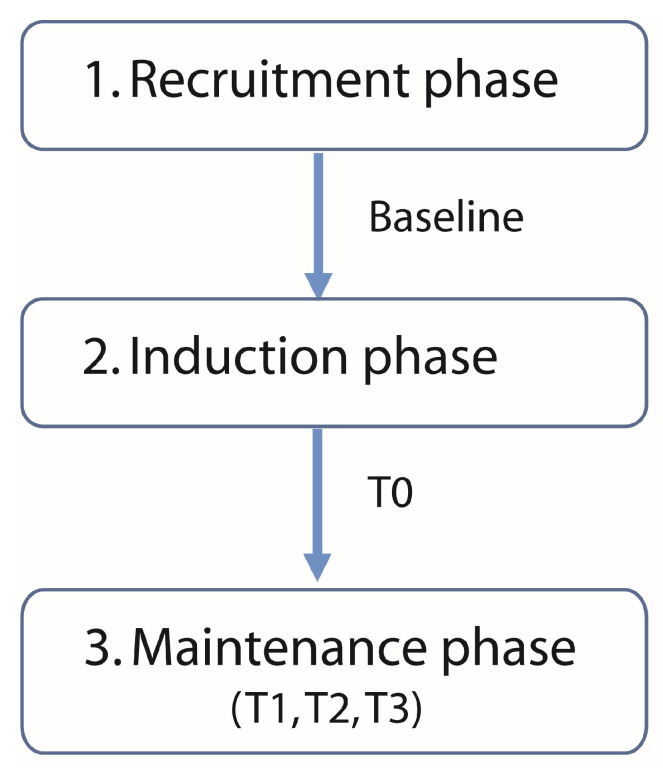
Protocol was divided into 3 main phases: (1) recruitment, when patients were enrolled in the experiment; (2) induction, when the patients received 10 days of tDCS treatment; (3) maintenance, when patients received a session of tDCS every 2–3 sessions per month. The data for the analysis were separated into baseline (5 days before the induction phase), T0 (5 days after the induction phase), T1 (3 months after the induction phase), T2 (6 months after the induction phase), T3 (12 months after the induction phase).

**Figure 2 biomedicines-12-00115-f002:**
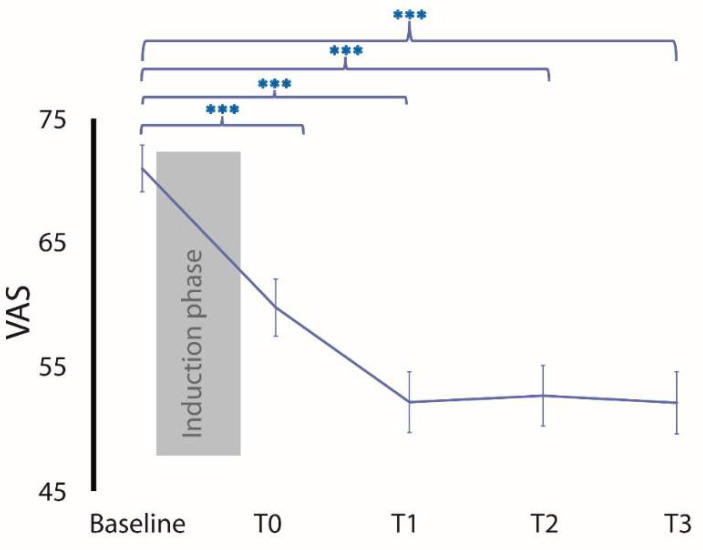
Timeline of “*responder*” group over 12 months. Grey area corresponds to the induction phase (10 days of tDCS). There was a significant decrease in pain intensity (VAS) over time after the induction phase. Effect persisted until T3. (Curly bracket refers to Conover’s post hoc test.) Values are shown as the mean  ±  standard error of the mean, *** *p*  <  0.001.

**Table 1 biomedicines-12-00115-t001:** Clinical and demographic characteristics of outpatients.

Subjects (N)	155
**Gender (*males N (%)*)**	60 (38.7%)
**Age, years (*mean ± SD*)**	49.26 ± 11.69
**Duration of Pain, years (*median [IQR]*)**	6 [3–12]
**MMSE (*median [IQR]*)**	29 [27–30]
**BDI (*median [IQR]*)**	19 [12–28]
**VAS_Baseline, mm (*median [IQR]*)**	77 [62.5–86.5]
**Diagnostic Groups (N (%))**	
SCI	51 (32.9%)
FBSS	15 (9.7%)
FM	40 (25.8%)
Others	49 (31.6%)
**Baseline Medication (N (%))**	
Antidepressant	90 (58.1%)
Gabapentin	20 (12.9%)
Pregabalin	7 (4.5%)
Other Anticonvulsant	68 (43.8%)
Opioids (Minor and Major)	57 (36.7%)
Cannabinoid	2 (1.3%)
Benzodiazepine	62 (40%)
NSAID	25 (16.1%)
Other Analgesic	28 (18%)
Baclofen	17 (10.9%)
Capsaicine/Local Administration	8 (5.2%)
Other Medications	58 (37.4%)

Abbreviations: N, number of patients; SD, standard deviation; IQR, interquartile range; mm, millimeters; MMSE, Mini-Mental Status Examination; BDI, Beck Depression Inventory; VAS, Visual Analogue Scale; SCI, spinal cord injury; FBSS, failed back surgery; FM, fibromyalgia; NSAID, non-steroidal anti-inflammatory drugs.

**Table 2 biomedicines-12-00115-t002:** Clinical and demographic characteristics of outpatients by diagnostic group.

N	SCI (51)	FBSS (15)	FM (40)	OTHERS (49)	*p*
**Gender (*male* (%))**	68.6%	33.3%	7.5%	34.6%	***p* < 0.001**
**Age, years (*mean* ± SD)**	48.2 ± 11.3	50.5 ± 9.0	48.8 ± 9.2	50.4 ± 14.5	*p* = 0.780
**Duration of Pain, years (*median [IQR]*)**	6 [2–12]	6 [4–13]	10 [6–17]	4 [2–7]	***p* < 0.001**
**MMSE (*median [IQR]*)**	28 [27–30]	29 [28–30]	29 [27–30]	29 [28–30]	*p* = 0.470
**BDI (*median [IQR]*)**	16 [11–25]	21 [16–27]	26 [16–34]	18 [10–26]	***p* < 0.003**
**VAS_Baseline, mm (*median [IQR]*)**	77 [64–87]	84 [76–92]	76 [60–88]	74 [59–81]	*p* = 0.168

Results for univariate test: Chi-square test, *one-way* ANOVA or Kruskal–Wallis test. Significant *p*-values are in bold. N, number of patients; SD, standard deviation; IQR, interquartile range; VAS, Visual Analogue Scale; mm, millimeters; SCI, spinal cord injury; FBSS, failed back surgery; FM, fibromyalgia.

## Data Availability

The data presented in this study are available on request from the corresponding author.

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
