# Peer review of "A Retrospective Study on tDCS Treatment in Patients with Drug-Resistant Chronic Pain"

_biomedicines, 2024, doi:10.3390/biomedicines12010115_

Round 1
Reviewer 1 Report
Comments and Suggestions for Authors
The paper is interesting and within the scope of journal. It is also well written and clear, but some improvements can be suggested to authors:
* Try to avid The Saxon Genitive and dashes in the paper title.
* Please avoid first person expressions in the paper (we, our) and use passive voice instead.
* Consider increasing number of keywords to at least 4, and trying to avoid using (mainly) abbreviations as keywords.
* Please avoid multiple referencing (e.g. [10-12]) which prevents readers from precisely identifying claims.
* Consider reformulating main objectives in past tense, stating what has been done and expressing novelty.
* In Table 1 caption there is no abbreviation for Range.
* Avoid starting sentences with abbreviation (e.g. begining of Section 2.2).
* Title of sections 3.1 and 3.2 end with colon for no apparent reason.
* It might be better to include Section 5 in Section 4, i.e. to transform 5. to 4.4.
* Section 6 needs to be expanded, to better and clearer include all findings of the study.
Comments on the Quality of English Language
Only moderate editing is needed, mainly for style (first person expressions, sentences start with abbreviations, etc.).
Reviewer 2 Report
Comments and Suggestions for Authors
The authors presented the results of continued research (Pérez-Borrego, Y.A.; Campolo, M.; Soto-León, V.; Rodriguez-Matas, M.J.; Ortega, E.; Oliviero, A. Pain Treatment Using TDCS in a Single Patient: Tele-Medicine Approach in Non-Invasive Brain Simulation. Brain Stimul 2014, 7, 334–335, 404 doi:10.1016/J.BRS.2013.11.008.; Camacho, A.B.; Borrego, Y.A.P.; Matas, M.J.R.; León, V.S.; Mateos, L.M.; Oliviero, A. Protocolo Terapéutico Del Dolor Con Técnicas de Estimulación No Invasiva. Medicine - Programa de Formación Médica Continuada Acreditado 2019, 12, 4451–4454, 407 doi:https://doi.org/10.1016/j.med.2019.03.026.) on transcranial direct current stimulation (tDCS) of the primary motor cortex (M1), which has been proven to produce a therapeutic effect better than a placebo in chronic pain. In the Abstract, they clearly formulated the aims to confirm the safety of one-year treatment, to describe the effects of tDCS on the pain intensity during one-year treatment, and to identify factors related to the success of treatment. For some incomprehensible reason, they also introduced the concept of patients treated effectively as "Respondents", creating confusion with the presentation of results in the following sentences, the content of which is far from clear. Moreover, the Study Design starts to be more confusing because the Authors in the same Abstract report on..."155 patients with pharmacological resistant chronic pain added tDCS treatment to their pharmacological treatment"..., which implies that it is difficult to evaluate the direct tDCS effect alone since patients could have been additionally treated pharmacologically. They admitted the weakness of the study in lines 362-366; moreover, patients were under different pharmacological treatments.
In the conclusion of the Abstract they confidentially stated that …”Anodal tDCS over M1 could be an adequate therapy (safe and efficient) to treat chronic pain since 58% of the patients are “Responders”. „… what is far from the clearly provided conclusion.
Reading the paper carefully I have found the study interesting and fruitful, but it needs better description starting from the Abstract and the Title which might better reflect the study content as follows: The retrospective study on tDCS treatment in patients with drug-resistant chronic pain.
The conclusion in Chapter 6 (lines 368-370) is very general and does not respond to the goals formulated for the study if the main method of the evaluation of the results was the Visual Analogue Scale reported with the surveys, which is known as very subjective. Considering the paper's content, I would be very careful with the statement: …” and clinicians and researchers can be trained easily.”… .
If authors included in their study the promise included in 4.3 Treatment success factors subsection as follows: ..."Of note, no other variables showed consistent and robust predictive value. In consequence, further studies will be necessary to clarify if pain intensity at the baseline can be associated with other variables to create a predictive model."..., I would agree with the confidence of the presented results.
Comments on the Quality of English LanguageModerate editing of English language and editorial corrections required
Round 2
Reviewer 2 Report
Comments and Suggestions for Authors
The authors responded well to most of my questions and substantive objections. The quality of the English language has not improved much; it's hard to talk about grammatical progress. For unknown reasons, the quality and method of presentation of the Tables has become different from that adopted in MDPI. However, after the above corrections to the manuscript, it could be qualified for publication in Biomedicines.
Some of the listed Refs. [i. e. 2, 3, 19] requires citation upgrading; most of them require editing corrections according to MDPI style.
Comments on the Quality of English LanguageThe quality of the English language has not improved much; it's hard to talk about grammatical progress.
